# Bacterial hopping and trapping in porous media

Tapomoy Bhattacharjee[1] & Sujit S. Datta [2]

Diverse processes—e.g. bioremediation, biofertilization, and microbial drug delivery—rely on bacterial migration in disordered, three-dimensional (3D) porous media. However, how pore-scale confinement alters bacterial motility is unknown due to the opacity of typical 3D media. As a result, models of migration are limited and often employ ad hoc assumptions. Here we reveal that the paradigm of run-and-tumble motility is dramatically altered in a porous medium. By directly visualizing individual *Escherichia coli*, we find that the cells are intermittently and transiently trapped as they navigate the pore space, exhibiting diffusive behavior at long time scales. The trapping durations and the lengths of "hops" between traps are broadly distributed, reminiscent of transport in diverse other disordered systems; nevertheless, we show that these quantities can together predict the long-time bacterial translational diffusivity. Our work thus provides a revised picture of bacterial motility in complex media and yields principles for predicting cellular migration.

[1] The Andlinger Center for Energy and the Environment, Princeton University, 86 Olden Street, Princeton, NJ 08544, USA. [2] Department of Chemical and Biological Engineering, Princeton University, 41 Olden Street, Princeton, NJ 08544, USA. Correspondence and requests for materials should be addressed to S.S.D. (email: ssdatta@princeton.edu)

While bacterial motility is well-studied in unconfined liquid media and at flat surfaces, in most real-world settings, bacteria must navigate heterogeneous three-dimensional (3D) porous media. For example, during an infection, pathogens squeeze through pores in tissues and biological gels, enabling them to spread through the body[1–6]. This process can also be beneficial; for example, a promising route toward cancer treatment relies on engineered bacteria penetrating into tumors and delivering anticancer agents[7,8]. In agriculture, the migration of rhizosphere bacteria through soil impacts crop growth and productivity[9–12], while in environmental settings, the process of bioremediation relies on motile bacteria migrating towards and degrading contaminants trapped in soils, sediments, and subsurface formations[13–15]. However, despite their potentially harmful or beneficial consequences, how motile bacteria move through 3D porous media remains completely unknown. As a result, our ability to accurately model migration in porous media is limited. This gap in knowledge hinders attempts to model the spread of infections, predict and control bacterial therapies, and develop effective agricultural and bioremediation strategies.

In free solution, peritrichous bacteria are propelled by a rotating bundle of flagella along ballistic runs of mean speed $\langle v_r \rangle$ and length $\langle L_r \rangle$; these are punctuated by rapid tumbles, arising when flagella spontaneously unbundle, that randomly reorient the cells. This run-and-tumble motion is thus diffusive over time scales larger than the run duration, with a translational diffusivity given by $\langle v_r \rangle \langle L_r \rangle / 3$[16]. How this behavior changes in a porous medium is unclear. A common assumption is that the bacteria continue to perform runs with a mean run speed $\langle v_r \rangle$, but with a shorter length $\langle L_r' \rangle < \langle L_r \rangle$ due to collisions with the solid matrix of the medium. These are thought to reorient the cells in a manner similar to tumbles, leading to a decreased diffusivity $\langle v_r \rangle \langle L_r' \rangle / 3$[17–21]. However, how to determine $L_r'$ is unclear, and in practice it is approximated by the average pore size or acts as an ad hoc parameter. Thus, while this approach is appealing due to its simplicity, it provides little fundamental understanding of how bacteria migrate in porous media. Indeed, the underlying assumption that the cells move via run-and-tumble motility has never been verified; typical 3D porous media are opaque, precluding direct observation of bacterial motion in the pore space.

Here, we report the first direct visualization, at single-cell resolution, of bacterial motion in 3D porous media. Our results overturn the assumption that the bacteria simply exhibit run-and-tumble motility with shorter runs; instead, we find a different form of motility in which individual cells are intermittently and transiently trapped as they move through the pore space. When a cell is trapped, it constantly reorients its body until it is able to escape; it then moves in a directed path through the pore space, a process we call hopping, until it again encounters a trap. We find that the distribution of hop lengths is set by the distribution of straight paths in the pore space, while the distribution of trapping durations shows power-law scaling similar to many other disordered systems, like amorphous electronic materials, colloidal glasses, and polymer networks. Remarkably, despite the heterogeneity of the pore space, we find that the mean hop length and trapping duration together can predict the long-time bacterial translational diffusivity. Our work thus provides a revised picture of bacterial motility in 3D porous media and yields principles for predicting cellular migration over large time and length scales.

## Results

**Directly visualizing bacterial motion in 3D porous media.** We prepare 3D porous media by confining jammed packings of ~10 µm-diameter hydrogel particles, swollen in liquid Lysogeny Broth (LB), in sealed chambers[22,23]. The internal mesh size of each particle is ~100 nm—much smaller than the individual bacteria, but large enough to allow unimpeded transport of nutrients and oxygen[24]. The packings therefore act as solid matrices with macroscopic interparticle pores[25] that bacteria can swim through (Fig. 1a). Importantly, because the hydrogel particles are highly swollen, light scattering from their surfaces is minimal. Our porous media are therefore transparent, enabling direct visualization of bacterial motility in the 3D pore space via confocal microscopy. This platform overcomes the limitation of typical media, which are opaque and thus do not allow for direct observation of bacteria within the pore space.

To tune the degree of pore confinement, we prepare four different media using different hydrogel particle packing densities. We characterize the pore size distributions of the media by dispersing $2 \times 10^{-3}$ wt% of 200 nm diameter fluorescent tracers in the pore space and tracking their thermal motion. Because the tracers are larger than the hydrogel mesh size, but are smaller than the inter-particle pores, they migrate through the pore space. A representative example of a tracer trajectory is shown in Fig. 1b; it reveals that the pore space is comprised of tortuous channels, each made up of a series of randomly-oriented, directed paths, similar to the pore space structure of many other naturally-occurring media[26]. Measuring the length scale at which

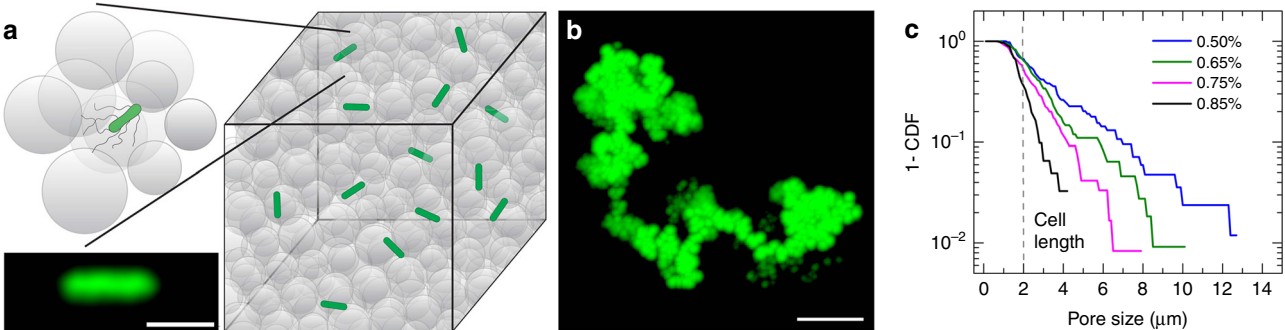

**Fig. 1** 3D porous media for direct visualization of bacteria motion. **a** Schematic of 3D porous media made by jammed packings of hydrogel particles, shown as gray circles, swollen in liquid LB medium. *E. coli*, shown in green, are dispersed at a low concentration within the pores between the particles. The packing is transparent, enabling imaging within the 3D pore space; inset shows a micrograph of a GFP-labeled bacterium. Scale bar represents 2 µm. **b** Time-projection of a 200 nm fluorescent tracer particle as it diffuses through the pore space, showing tortuous channels, each composed of a series of randomly-oriented, directed paths. Scale bar represents 5 µm. **c** Complementary cumulative distribution function 1-CDF of the smallest confining pore size *a* measured using tracer particle diffusion for four different porous media with four different hydrogel particle packing densities. Percentage indicates the mass fraction of dry hydrogel granules used to prepare each medium. Dashed line indicates cell body length of *E. coli* as a reference. Distributions are exponential, as indicated by straight lines on log-lin axes

the tracer mean-squared displacement (MSD) plateaus provides a measure of the smallest confining pore size $a$ of the medium (Supplementary Fig. 1). We plot 1-CDF($a$), where CDF($a$) = $\sum_0^a a\rho(a) / \sum_0^\infty a\rho(a)$ is the cumulative distribution function of measured pore sizes and $\rho(a)$ is the number fraction of pores having size $a$. Tuning the hydrogel particle packing density provides a way to tune the pore size distribution, with pores between 1 and 13 μm in the least dense medium, to pores between 1 and 4 μm in the densest medium (Fig. 1c). The pore sizes follow an exponential distribution for all four media, indicating a characteristic pore size (Supplementary Fig. 2); for simplicity, we refer to each medium by this characteristic size. Our hydrogel packings therefore serve as a model for many bacterial habitats, such as gels, soils, and sediments, which have heterogeneous pores ranging from ~1 to 10 μm in size, smaller than the mean bacterium run length and for many pores, smaller than the overall flagellum length ~7 μm[27–30].

In unconfined liquid, E. coli exhibit run-and-tumble motility. To quantify this behavior, we track the center $\vec{r}(t)$ of each individual cell with a time resolution of $\delta t = 69$ ms, projected in two dimensions, and analyze the time- and ensemble-averaged MSD, $\langle (r(t+\tau) - r(t))^2 \rangle$, as a function of lag time $\tau$. For short lag times, the MSD varies quadratically in time, indicating ballistic motion due to runs with a mean speed $\langle v_r \rangle = 28$ μm/s. By contrast, above a crossover time of ≈2 s, which corresponds to the mean run duration, the MSD varies linearly in time (red points, Fig. 2a). This transition to diffusive behavior is consistent with previous measurements[16].

We next investigate the influence of pore confinement on bacterial motion. We disperse the E. coli within the porous media at $6 \times 10^{-4}$ vol%, sufficiently dilute to minimize nutrient consumption and intercellular interactions. We track cell motion for at least 10 s, five times larger than the unconfined run duration, and focus our subsequent analysis on cells that exhibit motility within the tracking time. A mutant that cannot assemble flagella shows negligible motility, indicating that motion due to thermal diffusion and surface pili is insignificant (Supplementary Fig. 3). If pore confinement were to simply reduce the run length, as is often assumed, the MSDs would still exhibit a crossover between ballistic and diffusive motion, but at earlier lag times. We find markedly different behavior from this prediction. For short lag times, the MSDs vary as $\tau^{1.5}$, indicating superdiffusive motion. By contrast, above a crossover time $\tau_c$ (stars in Fig. 2a), the MSDs vary as $\tau^\nu$, where the exponent $0 < \nu \leq 1$ indicates subdiffusive behavior. Rescaling each MSD by its crossover point highlights these two regimes (Fig. 2b); moreover, it reveals that $\nu$ decreases with increasing pore confinement, approaching ≈0.5 for the densest medium. Analysis of the distribution of cell displacements at different lag times supports this finding (Supplementary Fig. 4). Our results thus contradict the idea that the paradigm of run-and-tumble motility persists in a porous medium.

**Bacteria move via intermittent hopping and trapping.** Close inspection of the individual cell MSDs reveals that the sub-diffusion is transient: at sufficiently long lag times, individual MSDs can again become diffusive (Supplementary Fig. 5), an

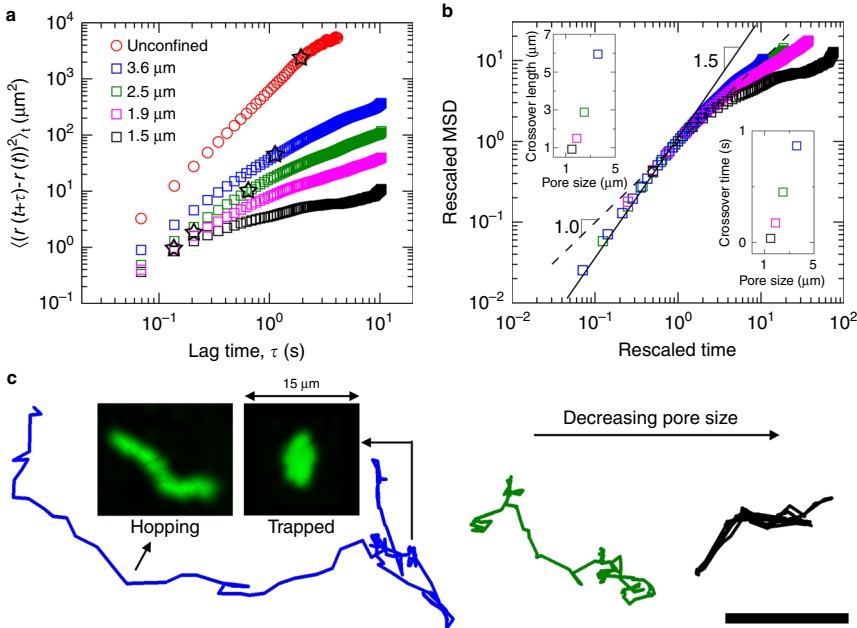

**Fig. 2** Bacteria move through 3D porous media via intermittent trapping and hopping. **a** Ensemble average mean-squared displacement (MSD) as a function of lag time for unconfined bacteria (red) and for bacteria in porous media with increasing amounts of confinement (blue, green, magenta, black). Stars indicate deviation from ballistic motion for unconfined bacteria, or deviation from superdiffusive motion for bacteria in porous media. Legend indicates characteristic pore sizes of the different media. **b** Rescaling by the crossover length and time scales (stars in **a**) indicates two regimes of motion for bacteria in porous media: superdiffusive motion at short times with the MSD scaling as $\tau^{1.5}$, and subdiffusive motion at long times with the MSD scaling as $\tau^\nu$ with the exponent $0 < \nu \leq 1$ decreasing with pore-scale confinement. This behavior is in stark contrast to simple run-and-tumble motion and instead reflects intermittent trapping of cells as they move. Insets show crossover lengths and times for different media; crossover lengths do not scale linearly with the measured characteristic pore sizes due to pore-size heterogeneity in the media. **c** Representative single-cell trajectories reveal switching between two modes of motion: hopping, in which bacteria move through extended, directed paths through the pore space, and trapping, in which bacteria are confined for extended periods of time. Insets show time projections of the cell body in the hopping and trapping state; trapped cells continue to reorient their bodies until they can escape and continue to hop through the pore space. Decreasing the pore size decreases the hop lengths, indicated by the green and black trajectories (characteristic pore sizes are 3.6, 2.5, and 1.5 μm from left to right). Scale bar represents 10 μm

effect that is masked in averaging. Such transient subdiffusion is known to arise from transient trapping within a heterogeneous environment[31–38]. Indeed, swimming bacteria are known to idle at solid surfaces and in tight spaces, slowing down or stopping altogether due to hydrodynamic and physicochemical interactions[39–41]. We thus propose that the individual cells hop through directed paths in the pore space, becoming transiently trapped when they encounter tight or tortuous spots, leading to the observed subdiffusive behavior. Careful inspection of the individual trajectories supports this hypothesis. We observe two distinct migration modes that the cells intermittently switch between (Supplementary Movie 1, Fig. 2c): hopping, in which a cell continuously moves through an extended, directed path through the pore space, and trapping, in which the cell is confined to a ~1 μm-sized region for up to ≈40 s. Moreover, as pore confinement increases, the hop lengths decrease, as exemplified by the different trajectories shown in Fig. 2c; this observation supports the idea that hops are guided by the geometry of the pore space.

To differentiate between hopping and trapping, we calculate the instantaneous speed of each cell as it moves through the pore space, $v(t) = |\vec{v}(t)| \equiv |\vec{r}(t + \delta t) - \vec{r}(t)|/\delta t$. The speeds are broadly distributed (Supplementary Fig. 6); however, the temporal trace of $v(t)$ exhibits the expected intermittent switching between fast hopping and slow trapping (Fig. 3a). We find similar motility behavior for all cells (Supplementary Fig. 7). We therefore define hops as intervals during which a cell moves faster than or equal to a threshold value of 14 μm/s, half the mean unconfined run speed $\langle v_r \rangle$. This definition corresponds to a hop

length of at least 1 μm, the smallest measured pore size, in each time step $\delta t$. Conversely, trapping is characterized by intervals during which a cell moves slower than the threshold $0.5\langle v_r \rangle$, or less than the smallest measured pore size in each time step. Importantly, our subsequent results do not appreciably change for different choices of the speed threshold up to $\langle v_r \rangle$ (Supplementary Fig. 8).

Our hypothesis suggests that a key difference between hopping and trapping is the directedness of the cell motion: during the course of a hop, a cell should maintain its direction of motion, while when trapped, the cell should constantly reorient itself until it can hop again (Fig. 2c). Indeed, the temporal trace of the velocity reorientation angle $\delta\theta(t) \equiv \tan^{-1}[\vec{v}(t) \times \vec{v}(t + \delta t)/\vec{v}(t) \cdot \vec{v}(t + \delta t)]$ also exhibits intermittent switching between hopping, with small $\delta\theta$ indicating directed motion, and trapping, with larger $\delta\theta$ indicating successive reorientations (Fig. 3b). We quantify this behavior by calculating the probability density $P(\delta\theta)$ for either hopping or trapping. Consistent with our expectation, $P(\delta\theta)$ is peaked at $\delta\theta = 0$ for hops, confirming that they are highly directed (squares, Fig. 3c). By contrast, $P(\delta\theta)$ is broadly distributed over a range of $\delta\theta$ for trapped cells (circles, Fig. 3c), indicating that their motion is randomly oriented (Supplementary Fig. 9).

We shed further light on this behavior by directly visualizing the flagella themselves (Supplementary Movie 2). During hopping, they form a rotating bundle that propels each cell along a directed path; the cell eventually stops moving, becoming trapped (Fig. 3d, first frame). However, the flagella continue to rotate as a bundle for ≈16 s, much longer than the mean

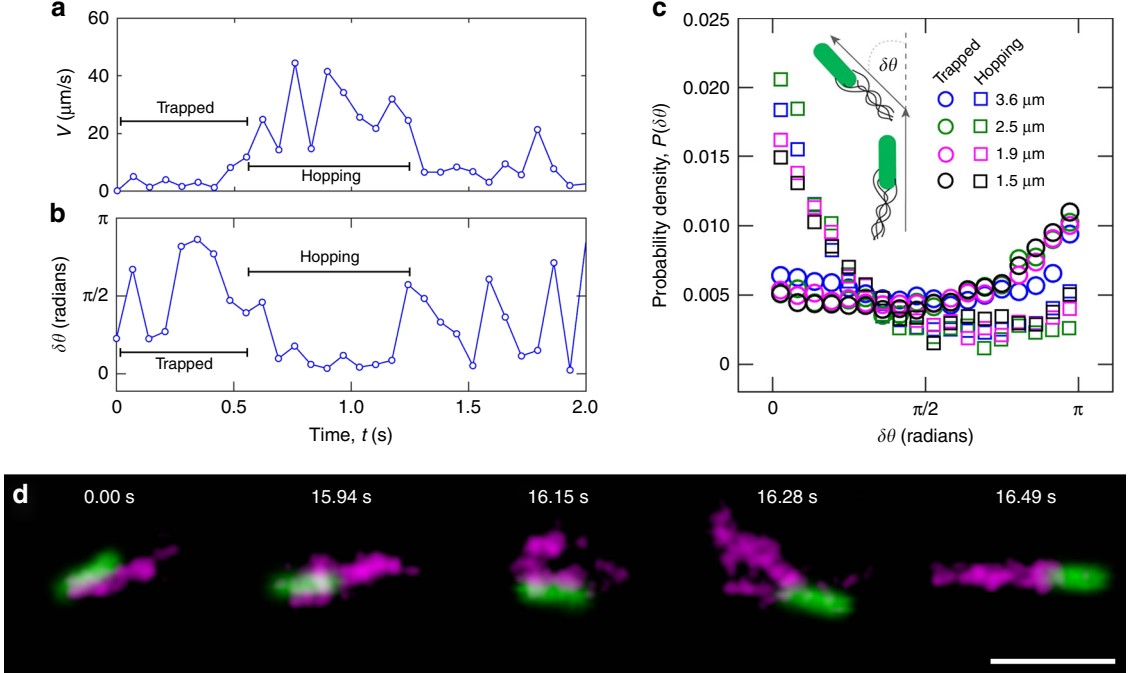

**Fig. 3** Properties of hopping and trapping of bacteria in 3D porous media. **a** The instantaneous speed of a representative cell as it moves through the pore space shows intermittent switching between fast hopping and slow trapping. The experimental uncertainty is smaller than double the symbol size, as described in Methods. **b** The cell velocity reorientation angle also exhibits intermittent switching between hopping, with small $\delta\theta$ indicating directed motion, and trapping, with larger $\delta\theta$ indicating successive reorientations of the cell body. The maximal experimental uncertainty is smaller than double the symbol size, as described in Methods. **c** Distribution of reorientation angle for all hops (squares) and all traps (circles); legend indicates characteristic pore sizes of the different media. $P(\delta\theta)$ of hops is peaked at $\delta\theta = 0$, indicating that hops are highly directed, while $P(\delta\theta)$ of traps is broadly distributed, indicating that motions of trapped cells are randomly oriented. **d** Direct labeling of flagella (magenta) shows that they remain bundled when a cell is trapped (first two frames), indicating that flagella unbundling is not required for trapping but instead that the cell has encountered a tight or tortuous spot. The cell body (green) continues to reorient itself, eventually enabling the flagella to unbundle (third frame) and re-bundle in a different orientation (fourth frame), and enabling the cell to continue to move through the pore space in a different direction (fifth frame). Scale bar represents 5 μm

unconfined run duration of 2 s (second frame); indeed, the longest run that we measure in bulk unconfined fluid is 5 s long, a factor of 3 shorter. Thus, flagellar unbundling—which leads to tumbling in unconfined media—is not required for cell trapping; instead, these measurements show that confinement can suppress unbundling, and trapping likely occurs when the cell encounters a tight or highly tortuous spot. The cell continues to reorient itself while trapped, eventually enabling the flagella to transiently unbundle (Fig. 3d, third frame) and re-bundle in a different configuration (Fig. 3d, fourth frame). This new flagellar configuration then enables the cell to escape its trap and continue to hop through the pore space in a different direction (Fig. 3d, fifth frame). We find similar behavior in another duplicate experiment: we again find that the flagella remain bundled during trapping, and the cell escapes its trap only when the flagella become transiently unbundled (Supplementary Movie 3).

**Statistics of hopping and trapping reflect the pore space disorder.** The pore space is heterogeneous; as a result, hopping and trapping are highly variable (Fig. 2c, Fig. 3a, b). We quantify this variability through the distributions of hop lengths $L_h$ and trapping durations $\tau_t$. For all media tested, both $L_h$ and $\tau_t$ are broadly distributed. The distributions of hop lengths show some overlap, likely reflecting the heterogeneity in the pore space; however, hops become shorter on average with increasing pore confinement, with a mean hop length of 3.24 μm for the least dense medium decreasing to a mean hop length of 2.14 μm in the densest medium (points in Fig. 4a). Interestingly, the probability density of trapping durations shows a power law decay over three decades in probability, with $\tau_t$ ranging from ≈0.4 to ≈40 s in our experiments (Fig. 4b); by contrast, the longest run that we measure in bulk unconfined fluid is nearly an order of magnitude shorter, and the measured hop durations are over an order of magnitude shorter. While the statistics are limited, the

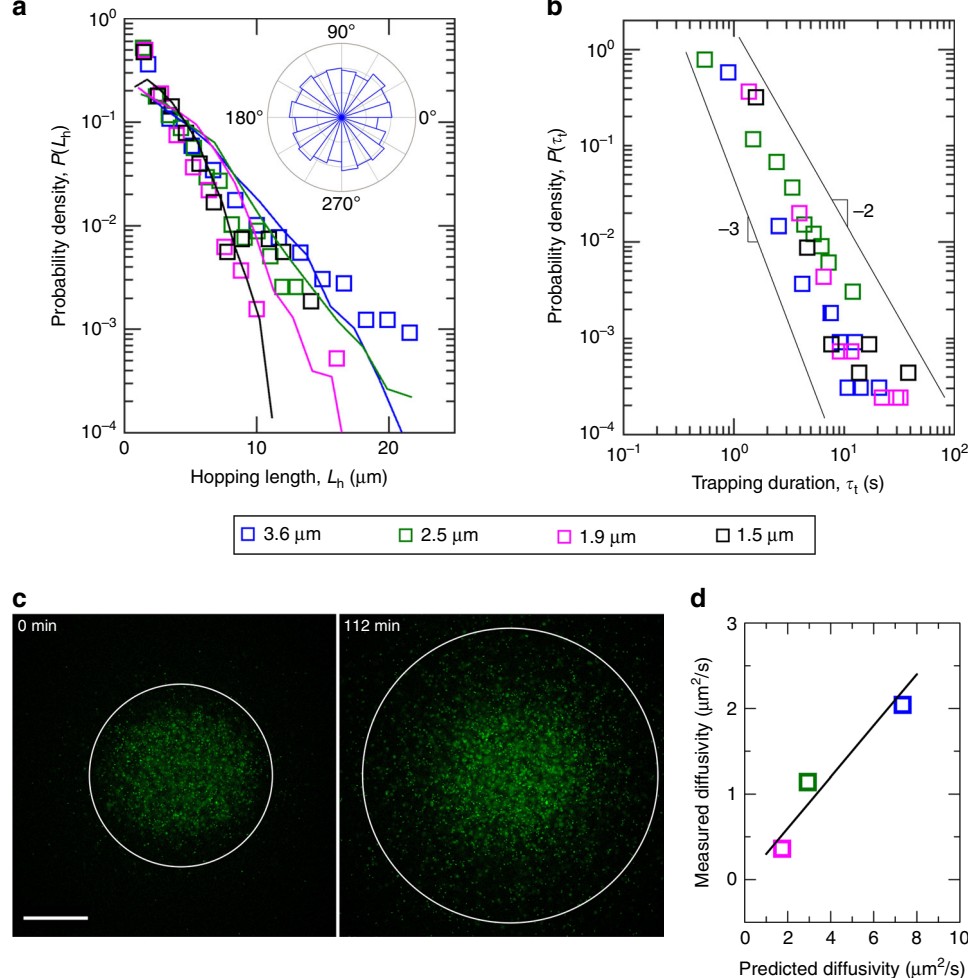

**Fig. 4** Measurements of hopping and trapping predict long-time translational diffusivity of bacteria. **a** Probability density of the measured hopping lengths for all hops; legend indicates characteristic pore sizes of the different media. The mean hop lengths are 3.24, 2.79, 2.14, and 2.14 μm from least dense to most dense medium. Curves show measured distribution of lengths of straight chords that can fit within the pore space. The agreement between the two indicates that hops are guided by the geometry of the pore space. Inset shows the distribution of hop orientations, indicating that hopping is completely random in space. **b** Probability density of the measured trapping durations for all traps; legend indicates characteristic pore sizes of the different media. We observe power-law scaling over three decades in probability characteristic of trapping in other disordered systems. **c** Confocal micrographs, taken 112 min apart, of a bolus of bacteria spreading within a 3D porous medium with characteristic pore size 3.6 μm; circle indicates the boundary of the bolus, determined using a threshold fluorescence intensity. Measuring the expansion of this boundary provides a way to directly quantify the long-time translational diffusivity due to cellular motility. Scale bar represents 250 μm. **d** Measurements of the long-time diffusivity agree with the prediction of a hopping-between-traps model. Points indicate three separate experiments in media with different pore sizes as indicated by the legend. Straight line indicates measured value = 0.3× predicted value

distributions of $\tau_t$ appear to scale as $\sim \tau_t^{-\alpha-1}$, with $\alpha$ decreasing weakly from $\approx 2$ to $\approx 1$ for increasing pore confinement (Fig. 4b). These results are insensitive to the choice of the minimum tracking duration (Supplementary Fig. 10). The measured power-law trapping durations are consistent with our measurements of transient subdiffusive followed by longer-time diffusive behavior, since the mean of the probability density function $P(\tau_t)$ is well-defined for the measured values of $\alpha$.

What determines the distribution of hop lengths? We expect that hops are guided by the geometry of the pore space itself: for a cell to move through the porous medium, it must be able to find a directed path. We therefore propose that the hop length distribution is given by the distribution of straight chords of length $L_h$ that can fit within the pore space, $f(L_h)$; this function is a fundamental metric in diverse problems involving directed transport, such as Knudsen diffusion, radiative transport, and fluid flow, in porous media[42]. We use our imaging of the pore space structure (Fig. 1b) to directly measure $f(L_h)$. The measured $f(L_h)$ are similar to the measured hop length distributions for all porous media tested, as shown in Fig. 4a, with broadly-distributed chord lengths that also become shorter with increasing pore confinement. This agreement confirms that hops are guided by the geometry of the pore space itself.

Our measurements of power-law distributed trapping durations (Fig. 4b) suggest that trapping is also determined by the disordered geometry of the pore space. Indeed, such distributions are a hallmark of disordered systems[43], arising for diverse examples including charge transport in amorphous electronic materials, macromolecule diffusion inside the cell, solute transport through porous media, molecular binding to and diffusion within membranes, and colloid diffusion through suspensions and polymer networks[34,36,38,44–46]. In all of these cases, the species being transported must hop through a disordered landscape of traps having varying confining depths[43]. Motivated by the striking similarities between the transport properties of other disordered systems and our measurements of sub-diffusive transport (Fig. 2), hopping and trapping (Fig. 3), and power-law trapping durations (Fig. 4b), we construct a phenomenological model of E. coli trapping within a porous medium. Our experiments demonstrate that a trapped cell constantly reorients itself until it can escape and continue to hop through the pore space (Fig. 3a, d). Inspired by previous work modeling the thermal diffusion of large polymers—which also must change configurations to escape traps—in random porous media[47–49], we thus propose that each trap can be thought of as an "entropic trap" characterized by a sharp depth $C$. This quantity is determined by the difference in the entropic contribution to the free energy between the trapped state and the transition state, in which a previously trapped cell can escape through an outlet[50]. It thus depends on the ratio between the number of possible ways a bacterium trapped in the pore can configure itself without being able to escape the trap, $\Omega_t$, and the number of possible ways the bacterium can configure itself at an outlet to escape the trap, $\Omega_e$, respectively; $\ln\Omega_t$ and $\ln\Omega_e$ thus represent the entropies of the trapped state and the transition state, respectively. For a given trap, the number of configurations $\Omega_t$ and $\Omega_e$, and thus $C$, likely depend on the pore size, the pore coordination number, and the size of the pore outlets; they also likely depend on the bacterium size and shape, flagellar properties, and any interactions with the pore surfaces. For a given porous medium with a broad distribution of traps, we then assume that the probability density of trap depths $C$ is given by $T(C) = C_0^{-1}e^{-C/C_0}$, similar to other disordered systems[43], where $C_0$ characterizes the average trap depth of the medium. We also assume that the probability for a cell to escape from a given trap of depth $C$ is given by an

Arrhenius-like relation, and thus, the trapping duration is given by $\tau_t = \tau_0 e^{C/X}$; $\tau_0$ is a characteristic time scale of swimming, while $X$ is an "activity" parameter that characterizes the ability of the cells to escape the traps. As such, this parameter likely depends non-trivially on the swimming speed, size and shape, flagellar properties, and surface properties of the cells; because our population is monoclonal, $X$ is a constant. The probability density of trapping durations is then given by $P(\tau_t) = \frac{T(C)}{d\tau_t/dC} = \frac{C_0^{-1}e^{-C/C_0}}{(\tau_0/X)e^{C/X}} = (\alpha\tau_0^\alpha)\cdot\tau_t^{-\alpha-1}$, where the parameter $\alpha \equiv X/C_0$ characterizes the competition between cellular activity and confinement in the porous medium. This scaling is consistent with our experimental measurements (Fig. 4b), with $\alpha$ decreasing weakly with increasing pore confinement. While a rigorous derivation is outside the scope of this work, this model suggests a tantalizing similarity between the motion of bacteria—which actively consume energy, and are thus out of thermal equilibrium—and a passive species navigating a disordered landscape.

**Hop lengths and trapping durations yield the long-time diffusivity**. Our measurements of hopping and trapping suggest a new way to calculate the long-time bacterial translational diffusivity. Because the pore space is disordered, the individual hop orientations are random (Fig. 4a, inset). We therefore model cell motion as a random walk for time scales longer than the mean trapping duration. Because $L_h \gg L_t$, we assume that the walk lengths are given by the hop lengths; however, because $\tau_t \gg \tau_h$, we assume that the walk times are given by the trapping durations, unlike a typical random walk. For simplicity, we use the ensemble-averaged values of $L_h$ and $\tau_t$; while this ansatz neglects variability in hopping and trapping, it provides a straightforward first step towards approximating the long-time diffusivity as $\approx \langle L_h^2\rangle/3\langle\tau_t\rangle$. We directly test this prediction by placing a spherical bolus of dilute cells within an initially cell-free porous medium with characteristic pore size $3.6\,\mu m$—for which we predict a diffusivity of $7\,\mu m^2/s$—and tracking radial spreading due to motility (Fig. 4c). We measure a diffusivity of $2\,\mu m^2/s$, comparable to the predicted value; by contrast, the run-and-tumble diffusivity with $L_r'$ given by the characteristic pore size is over one order of magnitude too large. Repeating this experiment for two different bacterial concentrations and at two other pore sizes yields similar agreement between the predicted diffusivity and the measured value in all cases (Fig. 4d, Supplementary Fig. 11); by contrast, the run-and-tumble diffusivity with $L_r'$ given by the characteristic pore size is always more than one order of magnitude too large. This agreement indicates that the migration of bacteria in a porous medium over large time and length scales can be explained by considering the dynamics of random hopping between traps.

## Discussion

Our experiments provide the first direct visualization of bacterial motion in 3D porous media. We find that bacteria do not simply exhibit run-and-tumble motility with runs shortened by confinement, as is commonly assumed. Instead, the individual cells hop through directed paths in the pore space, which are determined by the medium geometry, becoming transiently trapped when they encounter tight or tortuous spots. The distribution of trapping durations shows power-law scaling, revealing an unexpected analogy with transport in other disordered systems; rigorously determining how the dynamics of trapping depends on pore structure and surface properties, as well as cellular properties, will be a valuable direction for future work.

We observe that flagella remain bundled during cell trapping, indicating that pore-scale confinement can suppress unbundling. This suppression may be due to the flagella being excluded from a pore containing a trapped cell body, since many of the pores are smaller than the extended flagellum length in the porous media. We also observe that a cell leaves the trap only when its flagella become transiently unbundled; however, it may be possible that a cell could leave a trap without having to unbundle its flagella. For example, if a trapped cell continues to reorient its body in a pore, it could possibly leave the pore without having to unbundle its flagella once it meets an opening. We expect that the likelihood of this process depends on the geometry of the trapping pore—specifically, its size and the availability of pore "openings". We note that, conversely, each possible flagellar unbundling could enhance the body reorientation of a trapped bacterium, which would increase the probability that the cell finds the opening of the pore—suggesting a possible link between trapping and flagellar bundling. We speculate that the ability of the flagella to unbundle, which contributes to the ability of a cell to escape from a trap, depends on the interplay between the elasticity and geometry of the flagella and the size and geometry of the confining pores—which together determine the ability of flagella to deform—as well as hydrodynamic or chemical interactions with the pore surfaces, which are thought to suppress unbundling for the case of flat surfaces[51]. Elucidating this interplay will be an interesting direction for future work.

Previous studies of active particle motion in two-dimensional (2D) random media suggest that pore-scale confinement does not merely rescale long-time diffusive behavior, but fundamentally changes how active particles move;[52–55] our work provides an experimental complement to this body of work. Moreover, the revised picture of motility we present yields a way to predict the long-time bacterial diffusivity through a random walk model of hopping between traps. The diffusivity is central to describing cellular migration in settings ranging from infections, drug delivery, agriculture, and bioremediation; our results therefore have critical practical applications. More generally, the measured hopping and trapping process is reminiscent of a Lévy walk with rests[56], suggesting that bacteria swimming in a porous medium have unexpected similarities to migrating mammalian cells[57], robots searching for a target[58], and tracers in a chaotic flow[59].

## Methods

**Preparing 3D porous media**. To prepare the jammed hydrogel porous media we disperse dry granules of randomly crosslinked acrylic acid/alkyl acrylate copolymers (Carbomer 980, Ashland) directly into liquid LB media (2 wt% of Lennox Lysogeny Broth powder in DI water). We ensure a homogeneous dispersion by mixing this suspension for at least 12 h. Since the hydrogel is a cross-linked network of negatively charged polyelectrolytes, we finally adjust the pH to 7.4 by adding 10 N NaOH. This protocol results in a jammed, solid matrix of dense-packed hydrogel particles. We characterize the mechanical properties using rheology, and measure a linear shear modulus between 3 and 140 Pa for the packings. This modulus is ~$10^6$ times lower than the bacterial cell wall stiffness (~100 MPa), and the corresponding bulk modulus is ~$10^3$ times lower than the cell wall stiffness; we therefore do not expect that the media exert significant mechanical stresses on the bacteria.

We characterize the pore space structure by tracking dispersed 200 nm carboxylated polystyrene fluorescent nanoparticles (FluoSpheres, Invitrogen), which have a zeta potential (approximately −20 mV) comparable to those of *E. coli* (approximately −30 mV). We use an in-house custom MATLAB script to track the individual particles, identifying each tracer center using a peak finding function with subpixel precision and tracking its motion using the classic Crocker-Grier algorithm. For each tracer, we calculate its MSD as a function of lag time. For short lag times, the tracer diffuses unimpeded, and the MSD varies linearly in time. At longer times, the tracer becomes constrained by the surrounding solid matrix, and the MSD plateaus (dashed line in Supplementary Fig. 1). To calculate the smallest confining pore size $a$, we take the square root of this plateau value and add the tracer particle diameter.

**Bacterial culture**. Prior to each experiment, we prepare an overnight culture of *E. coli* (W3110) that constitutively expresses green fluorescent protein (GFP) throughout the cytoplasm at 30 °C and incubate a 1% solution of this culture in fresh LB for 3 h. At this point, the optical density is ~0.6. We then gently mix a small volume of this 0.6 OD culture in the hydrogel porous media to achieve a final bacterial concentration of 8000 cells/μL. This concentration is sufficiently dilute to minimize intercellular interactions, local gradients in oxygen or nutrient content, and changes in the overall concentration of oxygen and nutrients throughout the media; we do not detect any changes in cellular motility over the experimental time scale (~30 min), in agreement with this expectation. As a negative control, we test a strain containing a deletion of the flagellar regulatory gene *flhDC*, which does not assemble flagella; we detect negligible motility for this strain, indicating that our results probe motility due to flagellar bundling and rotation (Supplementary Fig. 3). For the experiment shown in Fig. 3d, we stain flagella using Alexa Fluor dye, washing away free dye before mixing the bacterial culture with the hydrogel porous medium.

**Tracking bacterial motion**. To monitor bacterial motility in 3D porous media, we confine 4 mL of the jammed hydrogel media containing bacteria in the bottom of a sealed glass-bottom petri dish (packing thickness ~1 cm) with an overlying thin layer (750 μl) of LB to prevent evaporation. We use a Nikon A1R inverted laser-scanning confocal microscope with a temperature-controlled stage at 30 °C to capture fluorescence images every 69 ms from an optical slice of 79 μm thickness. The images are captured at least 100 μm from the bottom of the container to avoid any boundary effects. We use an in-house custom MATLAB script to track the individual cells, identifying each cell center using a peak finding function with subpixel precision and tracking the cells using the classic Crocker-Grier algorithm. We track between 500 and 1500 cells for each porous medium tested. The imaging time scale for all experiments except those described in Fig. 4c (~1 min) is shorter than the cell division time, ensuring that our measurements of motility are not influenced by cellular growth and division. Over the experimental time scale (~30 min), using trapped cells as tracers of matrix deformations, we do not detect any changes in the pore structure of the packing due to evaporation, swelling, or microbial activity.

Our imaging yields a 2D projection of cell motion in 3D; our measurements therefore likely underestimate the cell speeds and hopping lengths, and likely overestimate the trapping durations. However, our measurements of hopping and trapping are robust to variations in the choices of the threshold speed cut off and the minimum tracking duration (Supplementary Figs. 8 and 10), suggesting that errors due to projection effects do not play an appreciable role. To further estimate the error due to 2D projection, we identify the polar angle below which any cells moving in 3D will be erroneously identified as trapped. Our threshold speed cut off to define a trapped cell corresponds to a maximum 2D displacement of ~ 1 μm per frame; thus, a cell would be erroneously considered to be trapped if it were moving out of the imaging plane at a polar angle smaller than $\theta^* = \tan^{-1}(1\,\mu m/39.5\,\mu m)$ from the vertical axis, where 39.5 μm is half the imaging slice thickness. The corresponding total solid angle is therefore $2 \times 4\pi\sin^2\theta^*$, where the factor of two accounts for both upward and downward motion. This solid angle is 0.13% of the total solid angle of the sphere, $4\pi$. Given that the measured velocities are isotropically oriented (Fig. 4a, inset), this estimate indicates that only 0.13% of cell motions are erroneously characterized due to 2D projection.

To determine the experimental uncertainty in the measured instantaneous speed $v(t) \equiv |\vec{r}(t + \delta t) - \vec{r}(t)|/\delta t$, we determine the uncertainty in our ability to measure the instantaneous position $\vec{r}(t)$ by tracking a completely immobile cell trapped in the densest porous medium. The MSD remains constant at 0.017 μm², corresponding to a positional uncertainty of $\Delta r = \sqrt{(0.017\,\mu m^2)} = 130$ nm. The uncertainty in $v$ is thus $2\Delta r/\delta t = 4$ μm/s, double the symbol size in Fig. 3a. To determine the corresponding uncertainty in the measured velocity reorientation angle $\delta\theta$, we calculate the maximal error in $\delta\theta$ for two consecutive velocity vectors $\vec{v}(t)$ and $\vec{v}(t + \delta t)$ arranged such that $\delta\theta = 0$, i.e., perfectly directed motion. The maximal uncertainty in $\delta\theta$ thus determined is ≈0.25 rads, smaller than double the symbol size in Fig. 3b.

**Measurement of chord length distribution**. To measure the chord length distribution $f(L_h)$, we construct maximum-intensity time-projections of movies of dispersed 200 nm fluorescent nanoparticles diffusing through the pore space. This provides a snapshot of the pore space geometry. We binarize these time projections into two phases—pore space and solid matrix—and lay randomly oriented lines across each image. A chord is defined as a line segment of length $L_h$ with every point in the pore space. We use these measurements to generate a discrete probability density function for each porous medium from all chord lengths, $f(L_h)$.

**Measurement of long-time diffusivity**. To measure the long-time translational diffusivity of bacteria within a porous medium (Fig. 4c), we premix a suspension of *E. coli* in a jammed medium of hydrogel particles to a final concentration of 8 million cells per μL. A small bolus (~190 nL) of this mixture is then injected inside an initially cell-free 0.5% jammed hydrogel porous medium and imaged using confocal microscopy. We quantify the radial spreading of this population due to cellular motility by taking a maximum-intensity time-projection (spanning 10 min)

both immediately after injecting the bolus and after 103 min. From the azimuthally averaged intensity profiles, we determine the position of the bolus boundary (circle in Fig. 4c) by defining a threshold intensity for the initial bolus and use this to measure the initial bolus radius, $R_0$. We then measure, for the spread bolus after $\Delta t = 103$ min, at what radius $R_0 + \Delta R$ a similar fluorescence intensity is observed; by tracking the bolus boundary, we avoid complications due to growth and division of trapped cells in the center of the bolus. Given that the spreading is isotropic in an effectively unbounded 3D medium, we approximate $\Delta R$ by the one-dimensional diffusion length, and the diffusivity is therefore given by $\frac{(\Delta R)^2}{4\Delta t} = \frac{(225\,\mu m)^2}{4(103\,min)} = 2\,\mu m^2/s$.

We expect that nutrient limitation does not play a role in these experiments. The overall change in the amount of nutrient levels is given by $\Delta C = k C_b (V_B/V)\Delta t$, where $k$ is the nutrient consumption rate per cell, $C_b$ is the bacterial concentration in the bolus used in our diffusivity measurements (corresponding to dilute cell volume fractions less than 0.6 vol% in our experiments), $V_B \approx 190$ nL is the bolus volume, $V \approx 25 \times 10^3 V_B$ is the total volume of the medium, and $\Delta t \sim 100$ min is the experimental time scale; the fractional change in nutrient is thus given by $\Delta C/C$, where $C$ is the initial dissolved nutrient concentration throughout the medium. As a representative example, we consider the consumption of oxygen or essential amino acids for E. coli (e.g. L-serine, L-aspartate). Using measured values of $C$ and $k$[60–62], we calculate fractional changes in nutrient levels smaller than 0.06% over the experimental timescale. We therefore expect that nutrient limitation does not play a role in our experiments.

We also expect that the spatial profile of nutrients experienced by single cells is uniform in our experiments. In the diffusion experiments, nutrient consumption by individual cells could generate cell concentration-dependent spatial inhomogeneities throughout the porous medium. For the range of concentrations explored here, inhomogeneities arising from nutrient consumption are rapidly homogenized by nutrient diffusion through the porous media, and thus, we do not expect cell concentration-dependent effects. This result can be seen via a calculation of the competition between nutrient consumption throughout a spherical bolus of cells and diffusion from the bolus boundary. The timescale of nutrient consumption is given by $C/kC_b$, and thus, the length scale over which the nutrient level varies is given by the diffusion length $2\sqrt{D(C/kC_b)}$, where $D$ is the diffusivity of single nutrient molecules. Because the hydrogel polymer is less than 1% of the total mass of the system, and the mesh size is ~100 nm (much larger than ~nm-sized single molecules), we assume that the hydrogel does not alter nutrient transport and availability. Using measured values of $D$[63,64], we calculate diffusion lengths of 2.0, 5.8, and 1.2 mm—three orders of magnitude larger than the size of a single bacterium—for the three representative examples of oxygen, L-serine, and L-aspartate. We therefore expect that the spatial profile of nutrients experienced by single cells is uniform in our experiments.

We also repeat this experiment four more times, testing bacterial concentrations of either 4–8 million cells per μL or 40–60 thousand cells per μL, and testing three different characteristic pore sizes $a = 3.6, 2.5, 1.9$ μm (Supplementary Fig. 11). Importantly, the measurements performed on populations of different concentrations within media of the same pore size yield comparable values of diffusivity, indicating that the measurements are independent of cell concentration and are indeed representative of non-interacting single cells.

## Data availability
The datasets generated during and/or analyzed during the current study are available from the corresponding author on reasonable request.

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

## Acknowledgements

It is a pleasure to acknowledge Tommy Angelini for providing microgel polymers; Average Phan and Bob Austin for providing fluorescent *E. coli*; Jeremy Cho for assistance with the diffusion analysis; and Tommy Angelini, Paulo Arratia, Bob Austin, Denis Bartolo, Alexander Berezhkovskii, Roseanne Ford, Henry Fu, Kela Lushi, Stanislav Shvartsman, Salvatore Torquato, Ned Wingreen, and Vasily Zaburdaev for stimulating discussions. This work was supported by start-up funds from Princeton University, the Project X Innovation fund, and a distinguished postdoctoral fellowship from the Andlinger Center for Energy and the Environment at Princeton University to T.B.

## Author contributions

T.B. and S.S.D. designed research; T.B. performed experiments; T.B. and S.S.D. analyzed data; S.S.D. developed the theory; T.B. and S.S.D. wrote the paper.

## Additional information

**Competing interests:** The authors declare no competing interests.

