## [Peer Review File · Nature Communications]

Reviewer #1 (Remarks to the Author):

This manuscript by Bhattacharjee and Datta reports an interesting experimental work on bacterial motions in porous media. They found that bacteria show a hopping and trapping behavior different from the classic run-and-tumble motion observed for bacteria in unconfined fluid. The hopping length distribution can be related to the pore size distribution of porous media, whereas the trapping time shows power-law scalings. The authors tried to draw an analogy to similar power-law trends occurring in other disordered systems. Based on these observations, they then predicted the long-diffusivity of bacterial suspensions in porous media and verified the results in experiments. I found the results are interesting and certainly deserve publication. But I do have several questions which should be addressed before the acceptance of the manuscript for Nature Communications.

- The authors compare the pore size and the pore size distribution with the body length of e coli. However, the flagella of bacteria is around 7 μm , significantly longer than the cell body and also the size of most pores in the porous media. This means that part of the flagella have to stay outside the pore when bacteria trap in the pores, which may be the reason why confinement suppresses the unbundling time. Can the authors comment on the relation between pore size and flagellar length and how it affects bacterial behaviors?

- Related to the above question, could the authors provide more quantitative evidence to support the claim that confinement suppresses unbundling? For example, it would be more convincing if the authors could show the distributions of run time with bundled flagella in confined samples versus those in the bulk fluid? It is also not clear to me how the reorientation of bacteria in pore enables the unbundling of flagella. If a trapped bacterium keeps rotating in the pore with bundled flagella, it can leave the pore once it meets an opening of the pore. Why is the process of flagellar bundling and unbundling important in this process?

- For Fig. 4A, it seems that, except the data of 3.6 μm , all the other data in different confinements overlap together. It is hard to see the trend claimed by the authors that with increasing pore confinement, the hop length becomes shorter. Similarly, I think the claim that there is "excellent" agreement between $f(L_h)$ and the hop length distributions is overstretched.

- In the phenomenological model, the authors proposed the distribution of "trap depths" follows an exponential decay. How do we understand the physical meaning of trap depths? Do they relate to the pore size, the number of outlets or the size of outlets of a pore? While I do like the analogy made by the authors to other disordered system, the phenomenological model proposed in the manuscript needs more justification in physical terms.

- Bacterial density used in the long-time diffusivity measurements was 10 times larger than that used in the single particle dynamics measurements. I am wondering whether density will influence the behaviors/diffusivity of bacteria. Can the authors measure the long-time diffusivity of bacteria in porous media using different bacterial concentrations and exam if there is any dependence on bacterial concentrations?

Reviewer #2 (Remarks to the Author):

In this paper, the authors use an innovative technique of placing run-and-tumble bacteria into a hydrogel environment, making it possible to image the motion of the bacteria through the confined medium. The authors find that the confinement changes the character of the motion from run-and-tumble to hopping and trapping. By analyzing the tracks of many bacteria, they show that the movement crosses over from ballistic at short time scales to subdiffusive at long time scales, with a subdiffusive exponent that decreases with increasing confinement. In contrast, unconfined bacteria show a crossover from ballistic motion at short times to purely diffusive motion at long times. The authors present evidence that individual bacteria continue to swim inside the pore but remain trapped until a reorientation allows the bacterium to escape into a new pore. A series of measurements support this picture. Use of the hydrogel to provide the porous environment is an important new advance that makes visualization, and detailed characterization, of the bacteria motion through a 3D porous medium possible for the first time. This is an important result that will be of great interest to the community, and I recommend it for publication after the authors address the comments below.

1. Although the motion of bacteria through 3D porous media was, up until this study, completely

unexplored, there has been some work on the motion of run-and-tumble bacteria through 2D random media which also shows that the medium does not merely rescale the long-time diffusive behavior but changes the character of the motion. The authors should mention some of this work:

C. Reichhardt et al, "Active matter transport and jamming on disordered landscapes," Phys. Rev. E 90, 012701 (2014).

C. Bechinger et al, "Active particles in complex and crowded environments," Rev. Mod. Phys. 88, 045006 (2016).

M. Zeitz et al, "Active Brownian particles moving in a random Lorentz gas," Eur. Phys. J. E 40, 23 (2017).

C.J.O. Reichhardt et al, "Avalanche dynamics for active matter in heterogeneous media," New J. Phys. 20, 025002 (2018).

2. Error bars, or some sense of the scale of error, in Fig. 3 (a,b) would be helpful.

3. The authors propose a simple trapping model to explain their data, and obtain phenomenological estimates of the trap depth. Since the bacteria must reorient in order to escape from the trap, I am wondering whether an entropic trap model might be a better fit. The concept of entropic trapping was first introduced in the context of long strands of DNA that needed to undergo conformational changes in order to pass through random porous media. The authors might consider adding a comment on this point.

Reviewer #3 (Remarks to the Author):

The authors report a study on the diffusive properties of bacterial suspension in 3D porous media formed by packed hydrogel microbeads. The porosity of the media was varied by changing the volume fraction of the microbeads and careful measurements of the dynamics of single bacteria were carried out. The motion of bacteria appears super-diffusive on the short time scale with a persistence length being proportional to the pore size, then it becomes sub-diffusive at intermediate times. Analysis of the time dependence of the individual bacterial speed reveals the existence of two distinctly different types of motion, described as hopping and trapping. The hopping length is found to be correlating with the pore size, but the trapping time depends weakly on the confinement. The statistics on the trapping time and hopping length conclude a revised picture of bacterial diffusivity in the porous media, suggesting the long-time diffusive behaviour is proportional to the square of the hopping length and inversely proportional to the trapping time since the bacteria is predominantly found in a trapped state. The proposed approximation has been experimentally tested for a one particular porosity revealing a qualitative agreement. The manuscript is well structured and clearly written. The results are presented logically. I believe, the findings will generate a considerable interest across interdisciplinary fields. I think, that the manuscript holds a sufficient quality and interest to warrant a publication in Nature Communications. However, in my opinion, the missing point of the study is the quantitative assessment of the long-time diffusivity at different pore size. Clearly, the MSD data is naturally limited by the bacteria confinement in 2D observation plane, hence the evaluation of long-time diffusivity in the given system is a serious challenge. However, the authors could possibly repeat the macroscopic measurements (Fig. 4C) of the diffusion coefficient at different porosities in order to confirm the proposed approximation. I would be willing to recommend the manuscript for publication once the authors evaluate the long-time diffusivity systematically by conducting additional measurements for an extended range of the pore sizes.

We would like to thank all three Reviewers for taking the time to carefully read the manuscript. We are gratified that all three Reviewers confirm the novelty, importance, and broad interest of our work, and that they all find our work of suitable quality for publication subject to some revisions. We greatly appreciate their insightful feedback, which has guided us to clarify the discussion and presentation of our results, and has encouraged us to further strengthen the manuscript by adding more experimental data. We believe that with the improvements we have made, the manuscript is now suitable for publication in *Nature Communications*. Thank you for your consideration.

Reviewer 1

This manuscript by Bhattacharjee and Datta reports an interesting experimental work on bacterial motions in porous media. They found that bacteria show a hopping and trapping behavior different from the classic run-and-tumble motion observed for bacteria in unconfined fluid. The hopping length distribution can be related to the pore size distribution of porous media, whereas the trapping time shows power-law scalings. The authors tried to draw an analogy to similar power-law trends occurring in other disordered systems. Based on these observations, they then predicted the long-diffusivity of bacterial suspensions in porous media and verified the results in experiments. I found the results are interesting and certainly deserve publication. But I do have several questions which should be addressed before the acceptance of the manuscript for *Nature Communications*.

We thank the Reviewer for carefully reading the manuscript, and are encouraged that the Reviewer finds the manuscript interesting and deserving of publication. We completely agree with all of the Reviewer's thoughtful questions and comments, and we have directly addressed them in the revised manuscript as described below. The Reviewer's feedback has guided us to clarify our discussion and to add more experimental data in the manuscript, which has strengthened the work. We are grateful to the Reviewer for providing such constructive feedback.

1) The authors compare the pore size and the pore size distribution with the body length of *e. coli*. However, the flagella of bacteria is around 7 μm , significantly longer than the cell body and also the size of most pores in the porous media. This means that part of the flagella have to stay outside the pore when bacteria trap in the pores, which may be the reason why confinement suppresses the unbundling time. Can the authors comment on the relation between pore size and flagellar length and how it affects bacterial behaviors?

The Reviewer raises an excellent point. The Reviewer is correct that the bacterial flagella are $\sim 7\mu\text{m}$ long, larger than the cell body as well as larger than the measured size of most pores in our media. Indeed, our ability to investigate bacterial motility in such pore-scale confinement is a key strength of our work; our experimental platform serves as a model for many real-world bacterial habitats (e.g. gels, soils, and sediments), which also have pores ranging from ~ 1 to $10\mu\text{m}$ in size. A unique result of this pore-scale confinement is that some of the flagella have to stay outside a pore when it traps a single cell, as the Reviewer correctly points out. We thank the Reviewer for this useful comment, and have now explicitly clarified the discussion of the geometric features characterizing pore-scale confinement in the revised manuscript.

We also fully agree with the Reviewer's insightful suggestion that flagellar exclusion from the pores may be the reason why confinement suppresses the unbundling time. We believe the Reviewer is completely correct, and are currently testing this intriguing hypothesis in follow-up work. Specifically, we hypothesize that the ability of flagella to unbundle, which contributes to the ability of bacteria to escape from a trap, depends on the interplay between the elasticity/geometry of the flagella and the size/geometry of the confining pores—which together determine the ability of flagella to deform—as well as hydrodynamic or chemical interactions with the pore surfaces, which are thought to suppress unbundling for the case of flat surfaces. Elucidating this physics will be an exciting and important extension of our work, and we expect that our results will motivate further research on the subject. We have now explicitly discussed this point in the revised manuscript. We thank the Reviewer for this comment, which has helped us clarify our discussion and has motivated ongoing work in our lab.

2) Related to the above question, could the authors provide more quantitative evidence to support the claim that confinement suppresses unbundling? For example, it would be more convincing if the authors could show the distributions of run time with bundled flagella in confined samples versus those in the bulk fluid? It is also not clear to me how the reorientation of bacteria in pore enables the unbundling of flagella. If a trapped bacterium keeps rotating in the pore with bundled flagella, it can leave the pore once it meets an opening of the pore. Why is the process of flagellar bundling and unbundling important in this process?

We appreciate the Reviewer's guidance in clarifying our discussion of the influence of confinement on flagellar unbundling. We have directly followed the Reviewer's recommendation and have included a direct comparison of the measured durations of bacteria running in unconfined bulk fluid and the duration of flagellar bundling in the revised manuscript. We have also performed an additional experiment in which we directly visualize a bacterium with labeled flagella in a porous medium; we again find that the flagella remain bundled during cell trapping, and the cell leaves the trap only when the flagella become transiently unbundled (Movie S3). Finally, guided by the Reviewer's questions, we have clarified the discussion and interpretation of our results, as well as our discussion of flagellar unbundling and its relation to bacterial trapping, in the revised manuscript. We thank the Reviewer for their questions, which have helped us to strengthen our discussion of flagellar unbundling.

The measurements shown in Figure 3D of our manuscript show an example of a bacterium that is trapped with its flagella bundled for ≈ 16 s in a porous medium; by contrast, the *longest* run that we measure in bulk unconfined fluid is 5 s long, a factor of 3 shorter. As the Reviewer suggests, this comparison of time scales, which we now include in the revised manuscript, provides quantitative evidence that pore-scale confinement can suppress unbundling. However, we also agree with the Reviewer that these measurements provide evidence that pore-scale confinement *can*—but may not always—suppress unbundling. Indeed, we hypothesize that the ability of flagella to unbundle depends on the interplay between the elasticity/geometry of the flagella and the size/geometry of the confining pores—which together determine the ability of flagella to deform—as well as hydrodynamic interactions with the pore surfaces, which are thought to suppress unbundling for the case of flat surfaces. Elucidating this physics will be an exciting and important extension of our work, and we expect that our results will motivate further research on the subject. We have now explicitly clarified this point in the revised manuscript.

We also fully agree with the Reviewer that while our experiment shows an example in which flagella remain bundled during cell trapping, and the cell leaves the trap only when the flagella become transiently unbundled, it is possible that a cell could leave a trap without having to unbundle its flagella. Specifically, we agree with the Reviewer that if a trapped bacterium keeps rotating in a pore, it could possibly leave the pore without having to unbundle its flagella once it meets an opening. This process likely depends on the geometry of the confining pore—specifically, its size and the availability of pore “openings”—and determining the conditions under which it occurs will be an interesting direction for future work. We note that, conversely, each possible flagellar unbundling could enhance the body reorientation of a trapped bacterium, which would increase the probability that the cell finds the opening of the pore—suggesting a possible link between trapping and flagellar bundling. We have now explicitly clarified these points in the manuscript. Moreover, we note that following the Reviewer's suggestion, we have performed an additional experiment in which we directly visualize a bacterium with labeled flagella in a porous medium; we again find that the flagella remain bundled during cell trapping, and the cell leaves the trap only when the flagella can become transiently unbundled (shown in Movie S3 of the revised manuscript).

The Reviewer makes an excellent suggestion to compare the measured durations of bacteria running in unconfined bulk fluid and hopping/trapping in porous media. We have directly followed this suggestion and now present these data in the revised manuscript. We note that hops in porous media are frequently shorter than runs in bulk fluid, due to the tortuosity of the pore space: long, directed paths are less frequent in confining porous media, as quantified by the distributions in Figure 4A. However, as the Reviewer suggests, we find that bacterial trapping in porous media—which our direct visualization suggests is correlated with flagella remaining bundled—can span much longer durations than the longest runs in bulk fluid. Specifically, we find that individual bacteria remain trapped in porous media for

up to 40 s in the case of the smallest-pore media (1.5 μm pore size) and up to 22 s in the case of the largest-pore media (3.6 μm pore size); by contrast, the longest run that we measure in bulk unconfined fluid is 5 s long, nearly an order of magnitude shorter. Together, these data provide more quantitative evidence suggesting that pore-scale confinement can indeed suppress unbundling. We thank the Reviewer for encouraging us to analyze these data, which we now include in the revised manuscript.

We thank the Reviewer for these insightful and thoughtful questions. As described above, we have followed the Reviewer's recommendations in all cases, performed further experiments and data analysis, and have clarified the presentation of our results in the revised manuscript—which together strengthen our discussion of the interplay between pore-scale confinement, trapping, and flagellar bundling as revealed by our experiments.

3) For Fig. 4A, it seems that, except the data of 3.6 μm , all the other data in different confinements overlap together. It is hard to see the trend claimed by the authors that with increasing pore confinement, the hop length becomes shorter. Similarly, I think the claim that there is "excellent" agreement between $f(L_h)$ and the hop length distributions is overstretched.

We fully agree with the Reviewer, and we appreciate their help in improving our discussion of these data. The Reviewer is completely correct that the distributions of hop lengths show some overlap, likely reflecting the heterogeneity in the pore space; we have now clarified this point in the revised manuscript. However, we note that the mean hop lengths decrease with decreasing pore size: we measure mean hop lengths of 3.24, 2.79, 2.14, and 2.14 μm for media with characteristic pore sizes given by 3.6, 2.5, 1.9, and 1.5 μm . We now include these data in the revised manuscript. We also agree that the claim of "excellent" agreement was overstated, and have now rephrased this discussion in the manuscript.

4) In the phenomenological model, the authors proposed the distribution of "trap depths" follows an exponential decay. How do we understand the physical meaning of trap depths? Do they relate to the pore size, the number of outlets or the size of outlets of a pore? While I do like the analogy made by the authors to other disordered system, the phenomenological model proposed in the manuscript needs more justification in physical terms.

We are encouraged that the Reviewer likes the analogy we make to transport in other disordered systems, and we appreciate the opportunity to clarify the phenomenological model. Guided by the Reviewer's questions, and guided by Reviewer 2's suggestion, we have now provided a more extensive discussion of the underlying physics in the revised manuscript. Specifically, we propose that the traps can be thought of as "entropic traps", in which each trap depth C is determined by the difference in the entropic contribution to the free energy between the trapped state and the transition state, in which a previously trapped cell can escape through an outlet. This difference thus depends on the ratio between the number of possible ways a bacterium trapped in the pore can configure itself without being able to escape the trap, Ω_t , and the number of possible ways the bacterium can configure itself at an outlet to escape the trap, Ω_e , respectively; $\ln \Omega_t$ and $\ln \Omega_e$ thus represent the entropy of the trapped state and the transition state, respectively. We agree with the Reviewer's suggestion that Ω_t and Ω_e , and thus C , likely depend on the pore size, the pore coordination number, and the size of the pore outlets; they also likely depend on the bacterium size and shape, flagellar properties, and any interactions with the pore surfaces. Elucidating this physics will be an exciting and important extension of our work, and we expect that our results will motivate further research on the subject. The physical picture we propose of "entropic trapping" is inspired by previous work modeling the thermal diffusion of large polymers—which also must change configurations to escape traps—in random porous media. However, we emphasize that since bacteria actively consume energy, and are thus out of thermal equilibrium, more work is required to clarify the applicability of such a thermodynamic model. We expect that these exciting ideas will motivate future studies on this subject. We are grateful to the Reviewer for their questions, which encouraged us to expand and clarify our discussion of the trapping model. We have now included this expanded discussion in the revised manuscript.

5) Bacterial density used in the long-time diffusivity measurements was 10 times larger than that used in the single particle dynamics measurements. I am wondering whether density will influence the behaviors/diffusivity of bacteria. Can the authors measure the long-time diffusivity of bacteria in porous media using different bacterial concentrations and exam if there is any dependence on bacterial concentrations?

The Reviewer makes an excellent point, and we appreciate having the chance to improve our presentation of the diffusion measurements. We directly followed the Reviewer's suggestion and have now measured the long-time diffusivity of bacteria in porous media using two different bacterial concentrations and at three different pore sizes. Importantly, we find that measurements performed on populations ~100 times more dilute than our previous measurements yield comparable values of diffusivity, indicating that our measurements are indeed representative of non-interacting single cells as suggested in the manuscript. These additional data—which are now provided in Figure S11 of the revised manuscript—strengthen our manuscript considerably, and we thank the Reviewer for encouraging us to perform these experiments.

We also include calculations to estimate the possible influence of concentration on the bacterial environment. One mechanism by which higher cell concentrations could alter diffusion is by reducing the overall availability of nutrients. The overall change in the amount of nutrient levels is given by $\Delta C = kC_b(V_B/V)\Delta t$, where k is the nutrient consumption rate per cell, C_b is the bacterial concentration in the bolus used in our diffusivity measurements (corresponding to dilute cell volume fractions less than 0.6 vol% in our experiments), $V_B \approx 190$ nL is the bolus volume, $V \approx 25 \times 10^3 V_B$ is the total volume of the medium, and $\Delta t \sim 100$ min is the experimental time scale; the fractional change in nutrient is thus given by $\Delta C/C$, where C is the initial dissolved nutrient concentration throughout the medium. As a representative example, we consider the consumption of oxygen or essential amino acids for *E. coli* (e.g. L-serine, L-aspartate). Using measured values of C and k , we obtain fractional changes in nutrient levels smaller than 0.06% over the experimental timescale. We therefore expect that nutrient limitation does not play a role in our experiments.

Another mechanism by which higher cell concentrations could alter diffusion is via nutrient consumption, which could generate cell concentration-dependent spatial inhomogeneities throughout the porous medium. Indeed, we have found interesting concentration-dependent effects on cellular motility in porous media at cell concentrations much larger than those explored in the current manuscript, and are preparing a separate manuscript reporting those findings. For the range of concentrations explored in the current manuscript, inhomogeneities arising from nutrient consumption are rapidly homogenized by nutrient diffusion through the porous media, and thus, we do not expect cell concentration-dependent effects. This result can be seen via a calculation of the competition between nutrient consumption throughout a spherical bolus of cells and diffusion from the bolus boundary. The timescale of nutrient consumption is given by C/kC_b , and thus, the length scale over which the nutrient level varies is given by the diffusion length $2\sqrt{D(C/kC_b)}$, where D is the diffusivity of single nutrient molecules. Because the hydrogel polymer is less than 1% of the total mass of the system, and the mesh size is ~100nm (much larger than ~nm-sized single molecules), we assume that the hydrogel does not alter nutrient transport and availability, as found in previous work (e.g. Ref 22 of the manuscript). Using measured values of D , we obtain diffusion lengths of 2.0, 5.8, and 1.2 mm—three orders of magnitude larger than the size of a single bacterium—for the three representative examples of oxygen, L-serine, and L-aspartate. We thus expect that the spatial profile of nutrients experienced by the cells is uniform in our experiments.

We have now included the additional diffusivity measurements at different cell concentrations and pore sizes, and the expanded discussion describing the influence of cell concentration, in the revised manuscript. We thank the Reviewer for this question, which has strengthened our presentation of the diffusivity measurements.

Reviewer 2

In this paper, the authors use an innovative technique of placing run-and-tumble bacteria into a hydrogel environment, making it possible to image the motion of the bacteria through the confined medium. The authors find that the confinement changes the character of the motion from run-and-tumble to hopping and trapping. By analyzing the tracks of many bacteria, they show that the movement crosses over from ballistic at short time scales to subdiffusive at long time scales, with a subdiffusive exponent that decreases with increasing confinement. In contrast, unconfined bacteria show a crossover from ballistic motion at short times to purely diffusive motion at long times. The authors present evidence that individual bacteria continue to swim inside the pore but remain trapped until a reorientation allows the bacterium to escape into a new pore. A series of measurements support this picture. Use of the hydrogel to provide the porous environment is an important new advance that makes visualization, and detailed characterization, of the bacteria motion through a 3D porous medium possible for the first time. This is an important result that will be of great interest to the community, and I recommend it for publication after the authors address the comments below.

We thank the Reviewer for carefully reading the manuscript, and are encouraged that the Reviewer finds our experiments innovative and important, novel, and of broad interest, and recommends our manuscript for publication after addressing their comments. We completely agree with all of the Reviewer's thoughtful comments, and we have directly addressed them in the revised manuscript as described below. The Reviewer's feedback has guided us to clarify our discussion and the presentation of our results, which has strengthened the work. We are grateful to the Reviewer for providing such constructive feedback.

1) Although the motion of bacteria through 3D porous media was, up until this study, completely unexplored, there has been some work on the motion of run-and-tumble bacteria through 2D random media which also shows that the medium does not merely rescale the long-time diffusive behavior but changes the character of the motion. The authors should mention some of this work:

C. Reichhardt *et al*, "Active matter transport and jamming on disordered landscapes," *Phys. Rev. E* 90, 012701 (2014).
C. Bechinger *et al*, "Active particles in complex and crowded environments," *Rev. Mod. Phys.* 88, 045006 (2016).
M. Zeitz *et al*, "Active Brownian particles moving in a random Lorentz gas," *Eur. Phys. J. E* 40, 23 (2017).
C.J.O. Reichhardt *et al*, "Avalanche dynamics for active matter in heterogeneous media," *New J. Phys.* 20, 025002 (2018).

We fully agree with the Reviewer and we thank them for correcting our oversight. The Reviewer is completely correct that previous work on run-and-tumble motion through 2D random media also shows that the medium changes motility, and provides an important context to place our work within. We have now included this discussion and have cited these references in the revised manuscript, and thank the Reviewer for helping us to improve the discussion of our work.

2) Error bars, or some sense of the scale of error, in Fig. 3 (a,b) would be helpful.

The Reviewer raises an important point, and we thank them for pointing out our previous omission of error bars. The experimental uncertainty in the measured instantaneous speed v is $\approx 4 \mu\text{m/s}$, smaller than double the symbol size in Fig. 3a, and the experimental uncertainty in the measured velocity reorientation angle $\delta\theta$ is ≈ 0.25 rads, smaller than double the symbol size in Fig. 3b. We have now included this information, as well as a detailed discussion of the source of experimental uncertainties, in the revised manuscript. We thank the Reviewer for their guidance in improving the presentation of our measurements.

3) The authors propose a simple trapping model to explain their data, and obtain phenomenological estimates of the trap depth. Since the bacteria must reorient in order to escape from the trap, I am wondering whether an entropic trap model might be a better fit. The concept of entropic trapping was first introduced in the context of long strands

of DNA that needed to undergo conformational changes in order to pass through random porous media. The authors might consider adding a comment on this point.

We fully agree with the Reviewer, and are grateful for their insightful suggestion, which helps to improve our discussion of the trapping model. Guided by the Reviewer's suggestion, and Reviewer 1's comments, we have now provided a more extensive discussion of the underlying physics in the revised manuscript. Directly following the Reviewer's suggestion, we now propose that the traps can be thought of as "entropic traps", in which each trap depth C is determined by the difference in the entropic contribution to the free energy between the trapped state and the transition state, in which a previously trapped cell can escape through an outlet. Similar to the previous work on DNA transport in random porous media mentioned by the Reviewer, this difference depends on the ratio between the number of possible ways a bacterium trapped in the pore can configure itself without being able to escape the trap, Ω_t , and the number of possible ways the bacterium can configure itself at an outlet to escape the trap, Ω_e , respectively; $\ln \Omega_t$ and $\ln \Omega_e$ thus represent the entropy of the trapped state and the transition state, respectively. We propose that Ω_t and Ω_e , and thus C , likely depend on the pore size, the pore coordination number, and the size of the pore outlets; they also likely depend on the bacterium size and shape, flagellar properties, and any interactions with the pore surfaces. Elucidating this physics will be an exciting and important extension of our work, and we expect that our results will motivate further research on the subject. Moreover, we note that while previous work on entropic trapping considered the thermal diffusion of long DNA—which also must change configurations to escape traps—in random media, since bacteria actively consume energy and are thus out of thermal equilibrium, more work is required to clarify the applicability of such a thermodynamic model. We expect that these exciting ideas will motivate future studies on this subject. We are grateful to the Reviewer for their suggestion, which guided us to expand and clarify our discussion of the trapping model. We have now included this expanded discussion in the revised manuscript.

Reviewer 3

The authors report a study on the diffusive properties of bacterial suspension in 3D porous media formed by packed hydrogel microbeads. The porosity of the media was varied by changing the volume fraction of the microbeads and careful measurements of the dynamics of single bacteria were carried out. The motion of bacteria appears super-diffusive on the short time scale with a persistence length being proportional to the pore size, then it becomes sub-diffusive at intermediate times. Analysis of the time dependence of the individual bacterial speed reveals the existence of two distinctly different types of motion, described as hopping and trapping. The hopping length is found to be correlating with the pore size, but the trapping time depends weakly on the confinement. The statistics on the trapping time and hopping length conclude a revised picture of bacterial diffusivity in the porous media, suggesting the long-time diffusive behaviour is proportional to the square of the hopping length and inversely proportional to the trapping time since the bacteria is predominantly found in a trapped state. The proposed approximation has been experimentally tested for a one particular porosity revealing a qualitative agreement.

The manuscript is well structured and clearly written. The results are presented logically. I believe, the findings will generate a considerable interest across interdisciplinary fields. I think, that the manuscript holds a sufficient quality and interest to warrant a publication in *Nature Communications*. However, in my opinion, the missing point of the study is the quantitative assessment of the long-time diffusivity at different pore size. Clearly, the MSD data is naturally limited by the bacteria confinement in 2D observation plane, hence the evaluation of long-time diffusivity in the given system is a serious challenge. However, the authors could possibly repeat the macroscopic measurements (Fig. 4C) of the diffusion coefficient at different porosities in order to confirm the proposed approximation. I would be willing to recommend the manuscript for publication once the authors evaluate the long-time diffusivity systematically by conducting additional measurements for an extended range of the pore sizes.

We thank the Reviewer for carefully reading the manuscript, and are encouraged that the Reviewer finds our experiments careful, our results of broad interest, and the manuscript well written and of sufficient quality to be published after more diffusivity measurements are presented. We have directly followed the Reviewer's suggestion, and have performed additional long-time diffusivity measurements for three different pore sizes. In all cases, we find good agreement (within a factor of ≈ 3) between the measured diffusivity and the predicted diffusivity. These additional data—which are now provided in Figure 4D and S11 of the revised manuscript—strengthen our manuscript considerably, and we thank the Reviewer for encouraging us to perform these experiments.

Reviewer #1 (Remarks to the Author):

I appreciate the authors' answers to my questions. I believe the quality of the manuscript has been greatly improved. Thus, I recommend the publication of the manuscript in Nature Communications.

Reviewer #2 (Remarks to the Author):

The authors have provided detailed replies and some additional experimental data to address the comments of the referees. These modifications have improved the presentation of these innovative results. I recommend the paper for publication without further changes.

Reviewer #3 (Remarks to the Author):

The authors have addressed the concerns raised by the referee, so I can now recommend the manuscript for publication.

Reviewer 1

I appreciate the authors' answers to my questions. I believe the quality of the manuscript has been greatly improved. Thus, I recommend the publication of the manuscript in *Nature Communications*.

Reviewer 2

The authors have provided detailed replies and some additional experimental data to address the comments of the referees. These modifications have improved the presentation of these innovative results. I recommend the paper for publication without further changes.

Reviewer 3

The authors have addressed the concerns raised by the referee, so I can now recommend the manuscript for publication.

We are gratified that all three Reviewers confirm that our manuscript is now suitable for publication in *Nature Communications*. We appreciate all the Reviewers' time, and their insightful questions and feedback, which greatly helped us to strengthen the manuscript.